# Hitchhikers' Guide to Masked Latent Semantic Modeling

## Abstract

Masked Latent Semantic Modeling (MLSM) is a pre-training objective which – in contrast to masked language modeling (MLM) – changes the objective of pre-training from the reconstruction of the exact word forms to their latent semantic properties (LSPs). The LSPs are determined by performing sparse coding based on the hidden token representations derived from an auxiliary model. In this paper, we identify and carefully evaluate previously unexplored important properties of MLSM pre-training. Based on the results of our rigorous experiments, we formulate a series of recommendations and best practices regarding MLSM pre-training for improving its efficiency. Among other recommendations, we propose a recipe for choosing the layer of the auxiliary model to determine the LSPs from, such that we can reduce the costs of pre-training MLSM pre-training, while maintaining or even surpassing the downstream fine-tuning capabilities of the resulting model. We also provide an improved implementation of MLSM, which reduces its computational requirements expressed in FLOPS by 33%. Besides the improved computational requirements, MLSM comes with better fine-tuning transferability, i.e., in our experience, the fine-tuning performance of MLSM pre-trained model checkpoints is on par or better than that of alternatively pre-trained models for twice the update steps. We release our code for reproducing our experiments at `github.com/[MASK]`

## 1 Introduction

Masked Latent Semantic Modeling (MLSM; Berend, 2023) has been recently proposed as a cognitively inspired alternative to masked language modeling (MLM) that relies on latent concepts inferred in an unsupervised manner. The core idea behind MLSM is that instead of requiring language models to output the exact identity of the masked words, they are expected to output a context-sensitive semantic characterization of the masked words in terms of their conceptual latent semantic properties as their pre-training task.

We illustrate the different kinds of outputs that models pre-trained with MLM and MLSM objective are expected to deliver in Figure 1. In case the masked token happens to be the word *dog*, the MLM pre-training loss is minimized if the model outputs *all* the probability mass to that exact token (Figure 1a).

This means that during the individual updates of MLM, we assume that for a particular masked token, there exist only a single unique correct substitute. As MLM pre-training progresses, language models become capable of outputting meaningful probability distributions for the substitutes of masked tokens, however, this requires vast amounts of diverse training data, something that is not as crucial for MLSM pre-training.

In contrast to MLM pre-training, the target output distribution of the model when using MLSM ranges over the latent semantic properties (abbreviated as LSP in Figure 1b) of the masked tokens. The LSPs used during MLSM pre-training are determined in an unsupervised manner from an auxiliary teacher model by expressing its hidden representations as a sparse linear combination of a codebook of semantic atoms.

Even though MLSM pre-trained models have been shown to outperform MLM pre-trained models regarding their transferability to downstream tasks, there are important open questions and practical considerations not discussed thoroughly enough, begging for further investigations and supportive empirical evidences. Our paper aims at answering those research questions via rigorous experiments for improving the understanding of MLSM pre-training. More specifically, we investigate the following research questions:

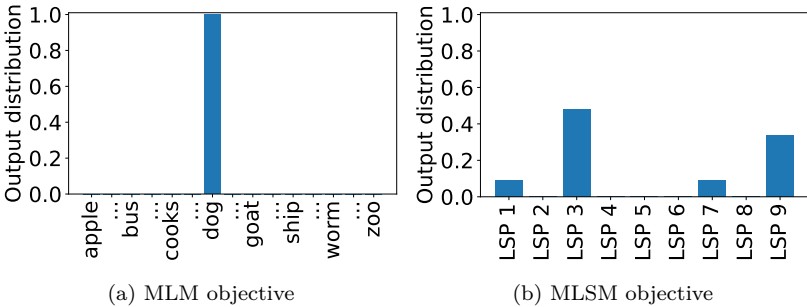

Figure 1: Comparisons of the probability distributions required by MLM (a) and MLSM (b) pre-training, with LSP referring to the latent semantic properties devised from the hidden representations of an auxiliary model in an unsupervised manner.

- how can we improve the efficiency of MLSM,
- what are the effects of determining LSPs on hidden representations that originate from different layers of the auxiliary model,
- how can we cheaply anticipate the fine-tuning transferability of an MLSM pre-trained model that is based on a particular layer of the auxiliary model,
- what are the effects of using different number of LSPs,
- how to test and improve linguistic capabilities of MLSM pre-trained models.

Throughout the paper, we highlight our key findings with colored background. We also release the pre-trained models that we created for our experiments at HuggingFace Hub, as well as our code base at github.com/[MASK].

## 2 Related work

Prior work suggests that sparse representations obtained from dense hidden vectors can convey the semantic properties of the words they describe (Berend, 2020; Yun et al., 2021). The core idea behind MLSM pre-training is that the latent semantic decomposition of the individual context-sensitive token representations can be exploited as a pre-training signal. Our paper advances knowledge on MLSM pre-training by providing a series of thorough experiments, related to various important implementation choices that has been previously overlooked in the literature. We also investigate the training costs and offer an improved implementation of MLSM.

MLSM also relates to prior research involving the integration of semantic categories into language model pre-training (Levine et al., 2020; Bai et al., 2022; Shani et al., 2023). The way MLSM progressed these approaches is that it determines the semantic properties to recover by the model in a context-sensitive manner, not requiring any external linguistic resources, such as WordNet (Fellbaum, 1998) or ConceptNet (Speer et al., 2017), hence it is capable of considering conceptual relations that go beyond hypernymy.

Since MLSM requires an auxiliary teacher model for determining the distribution of latent semantic properties of the masked tokens that the language model needs to recover as its pre-training task, it can be naturally framed as a special kind of knowledge distillation technique (Hinton et al., 2015; Aguilar et al., 2020). MLSM pre-training is also related to the line of research that incorporates alternative pre-training objectives as opposed to outputting the exact identity of the masked token (Levine et al., 2020; Yamaguchi et al., 2021; Alajrami & Aletras, 2022).

Despite the growing prominence of decoder-only generative large language models, encoder-based language models remain highly relevant in contemporary applications, such as retrieval-augmented generation (RAG). This is evidenced by the recent release of advanced encoder-based models like ModernBERT (Warner et al., 2024), NeoBERT (Breton et al., 2025) or EuroBERT (Boizard et al., 2025).

Our research also relates to research efforts that aim at enhancing the efficiency of pre-training for encoder-based language models (Izsak et al., 2021; Portes et al., 2023; Geiping & Goldstein, 2023). Our approach, however, differs from these earlier works, as they were focusing on architectural speedups and design choices of encoder models trained with the traditional pre-training paradigm, whereas we focus on modification of the learning objective in order to make pre-training more effective in terms of transferability to downstream tasks and better aligned with human perception.

Most recently, sparse autoencoders (SAEs) have became a popular tool for offering post-hoc interpretabiliy of LLM behavior (Huben et al. 2024; Lieberum et al. 2024; He et al. 2024; *inter alia*). This line of research relates to ours in that both involve sparse coding of neural activations, with the core difference being that we make use of the sparsified representations for improving the pre-training phase of encoder-based models. Concept features derived from a SAE have recently also been used for pre-training autoregressive large language models (Tack et al., 2025).

## 3 Masked Latent Semantic Modeling

We first overview how MLSM pre-training works, as it plays a central role in our experiments. A crucial difference in MLSM is that it does not output a distribution over the vocabulary of the model, but over the LSPs that are extracted from an auxiliary model as illustrated in Figure 1.

The way MLSM determines the LSP distribution of a token is by relying on an auxiliary model $\mathcal{T}$. In a preparatory phase, a sample of hidden representations produced by $\mathcal{T}$ is collected from its layer $l$ as $\{\boldsymbol{h}_1^{(l)}, \ldots, \boldsymbol{h}_N^{(l)}\}$. A dictionary learning problem (Mairal et al., 2009) is then solved of the form

$$\arg\min_{\boldsymbol{D}^{(l)}, \boldsymbol{\alpha}_j^{(l)} \in \mathbb{R}_{\geq 0}^k} \sum_{j=1}^N \frac{1}{2} \|\boldsymbol{h}_j^{(l)} - \boldsymbol{D}^{(l)}\boldsymbol{\alpha}_j^{(l)}\|_2^2 + \lambda\|\boldsymbol{\alpha}_j^{(l)}\|_1, \tag{1}$$

where $\boldsymbol{D}^{(l)} \in \mathbb{R}^{d \times k}$ is a dictionary matrix, with column vector norms bounded by 1, $\boldsymbol{\alpha}_j^{(l)} \in \mathbb{R}^k$ contains the sparse linear coefficients that indicate the extent to which the vectors from $\boldsymbol{D}^{(l)}$ are used in reconstructing the $d$-dimensional hidden representation from the $l$-th layer of $\mathcal{T}$, $\boldsymbol{h}_j^{(l)} \in \mathbb{R}^d$. $\lambda$ serves as a regularization coefficient, controlling for the level of sparsity in $\boldsymbol{\alpha}_j^{(l)}$.

Solving (1) is a one time effort, performed before the actual pre-training phase, having a negligible ($\ll 1\%$) computational overhead compared to the typical costs of pre-training. Once the dictionary matrix $\boldsymbol{D}^{(l)}$ is determined, it is used for determining the sparse contextualized representation for any $\boldsymbol{h}_i^{(l)}$, i.e., a hidden state from layer $l$ of $\mathcal{T}$ as

$$\arg\min_{\boldsymbol{\alpha}_i^{(l)} \in \mathbb{R}_{\geq 0}^k} \frac{1}{2} \|\boldsymbol{h}_i^{(l)} - \boldsymbol{D}^{(l)}\boldsymbol{\alpha}_i^{(l)}\|_2^2 + \lambda\|\boldsymbol{\alpha}_i^{(l)}\|_1. \tag{2}$$

Objective (2) is computationally convenient, as it does not require optimizing towards $\boldsymbol{D}^{(l)}$. With $\boldsymbol{D}^{(l)}$ being fixed from (1), obtaining the sparse linear coefficients $\boldsymbol{\alpha}_i^{(l)}$ constitutes an efficient to solve LASSO optimization problem.

We ensure in (2) that the coefficients of $\boldsymbol{\alpha}_i^{(l)}$ are all non-negative, but it is not guaranteed by default that it can be considered as a proper probability distribution. The non-negativity of $\boldsymbol{\alpha}_i^{(l)}$ allows us to conveniently turn it into a proper probability distribution via $\ell_1$-normalization. Intuitively, the $\ell_1$-normalized version of $\boldsymbol{\alpha}_i^{(l)}$ provides us the relative importance for each of the LSPs in the semantic characterization of the given token (producing distributions similar to the one in Figure 1b).

The probability distributions over the $k$ distinct LSPs are then used during the course of pre-training as the basis of calculating the loss function via the Kullback–Leibler divergence. That is, assuming that the distribution over the LSPs for a token is given by distribution $P$, and the output distribution provided by the network for the same token is given by distribution $Q$, the loss function is determined by $\sum_{j=1}^k P(j) \log \frac{P(j)}{Q(j)}$. We also provide a schematic overview of the MLSM pre-training in Figure 2.

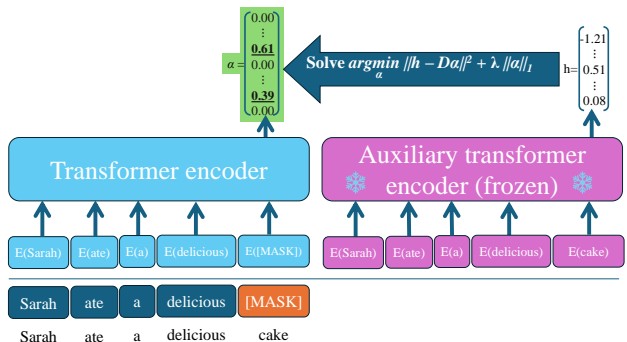

Figure 2: A schematic overview of MLSM pre-training.

### 3.1 Improving the efficiency of MLSM

Intuitively, MLSM pre-training may appear to be strictly more resource intensive than MLM pre-training, owing to the use of the auxiliary model $\mathcal{T}$. However, this is not necessarily the case, as the output space of an MLSM pre-trained model spans a considerably smaller domain compared to that of an MLM pre-trained model. Indeed, the output space of MLM models range over $V$, with $V$ being the entire vocabulary of the model, whereas MLSM models output a distribution over the $k$ LSPs, with $k \ll |V|$.

This means that for MLSM, the final unembedding layer – transforming the output of the final transformer block into model logits – can be of the shape $d \times k$ (as opposed to $d \times |V|$), where $d$ is the dimensions of the hidden state. The amount of FLOPS saved for producing model outputs outweigh the computational overhead of determining the target distribution of LSPs from the auxiliary model $\mathcal{T}$.

Berend (2023) mentions that "we introduce $k$ new special symbols into the output vocabulary of the model". That is, the number of parameters in the unembedding layer was $d \times (|V|+k)$, instead of $d \times k$. Additionally, in the official implementation of MLSM, an *unnecessary* full forward pass over $\mathcal{T}$ – including the calculation of the model logits – was conducted, even though MLSM requires only the hidden states of the auxiliary from its layer $l$. This means that for an auxiliary model $\mathcal{T}$, which uses hidden states of dimension $d_{\mathcal{T}}$ and consists of $L$ layers in total, the forward pass in $\mathcal{T}$ for layers $\{l+1, \dots, L\}$, as well as the computation related to the unembedding operation (involving multiplication with a $d_{\mathcal{T}} \times |V|$ matrix) can be spared, providing substantial improvement in the computational efficiency of MLSM pre-training.

## 4 Experiments

Throughout our experiments, we pre-trained multiple models that we evaluate over a diverse set of downstream tasks. We next describe the details of pre-training and evaluation involving both fine-tuning and zero-shot results.

### 4.1 Pre-training experiments

Unless stated otherwise, we relied on the `20230920` dump of the English Wikipedia as our pre-training corpus that we cleaned with the WikiBERT pre-processing pipeline (Pyysalo et al., 2021). During pre-training, we used the AdamW optimizer (Loshchilov & Hutter, 2019) with linear learning rate scheduling with a peak value of 1e−4, having a warm-up phase that constituted 2% of the total 100,000 update steps. Throughout pre-training, we employed an effective batch size of 1024 (using gradient accumulation of 8 with a batch size of 128) and a maximal sequence length of 128 subtokens.

We additionally ensured for better comparability between the different pre-training trials that the individual batches had identical contents and were processed in identical order. All together we conducted pre-training over approximately 102 million (100,000*1024) input sequences, spanning over nearly 13 billion tokens, i.e., 4 epochs over our 3.3 billion tokens pre-training corpus.

Unless stated otherwise, we created all our models from scratch using `DeBERTa-base` (He et al., 2021) backbone architecture. That is, our models consisted of 12 transformer blocks and employed a hidden vector of 768 dimensions. This configuration of DeBERTa comes with approximately 100 million model parameters, i.e., 8.3 million per encoder layer. Our MLSM models thus had substantially higher capacity compared to the models originally pre-trained with MLSM, i.e., Berend (2023) created 8 layer encoders with 512 dimensional hidden representations that resulted in approximately 25 million parameters related to the transformer blocks. Our experimental results hence allow us to assess if the benefits of MLSM pre-training also apply for models with substantially increased capacity.

Pre-training one model for our experiments took approximately 3 GPU days on a single NVIDIA A6000. This is in stark contrast to the computation involved in the creation of the official DeBERTa model that was trained for nearly 2,000 V100 GPU days (the 1 million update steps that they conducted took about 20 days, using 96 V100 GPUs (He et al., 2021)).

### 4.1.1 MLSM related hyperparameters

When performing MLSM, we relied on `bert-base` as the auxiliary model $\mathcal{T}$. Our auxiliary model – excluding its unembedding layer not required for producing the LSPs – consists of approximately 96 million parameters, but thanks to the improvements discussed in Section 3.1, we did not have to perform a full forward pass over the entire network for obtaining the LSP profile of the masked tokens.

The exact amount of computation required for determining the LSPs is dependent on hyperparameter $l$, i.e., the layer of $\mathcal{T}$ for extracting the hidden vectors from. (Berend, 2023) only considered using the hidden representations from the very last layer of $\mathcal{T}$ (i.e., $l = 12$), whereas in Section 4.2.1, we provide a series of controlled experiments on the role of altering the value of $l$.

We also investigate the effects on the choice of $k$, i.e., the number of LSPs to consider during MLSM pre-training. The values we checked were 1, 2, 4, 8 and 16 times the dimensionality of the hidden vectors employed in $\mathcal{T}$, meaning that we had $k \in \{768, 1536, 3072, 6144, 12288\}$. We report our results obtained with different choices of $k$ in Section 4.2.2. Unless stated otherwise, we employ $k = 3072$. For obtaining $\boldsymbol{D}^{(l)}$ according to (1), we relied on 2 million hidden vectors obtained from $\mathcal{T}$ produced on texts randomly sampled from our pre-training corpus.

### 4.1.2 Baseline models

We consider two baselines, the first being vanilla MLM pre-training, for which all overlapping hyperparameters were kept identical to those of MLSM pre-trained models. As MLSM uses a teacher–student paradigm, it is natural to compare to a knowledge distillation (KD) setting (Hinton et al., 2015; Aguilar et al., 2020).

During KD, we also rely on `bert-base-cased` as the auxiliary pre-trained teacher model $\mathcal{T}$. We use $\mathcal{T}$ to provide its predicted distribution for the substitutes of the masked tokens, that we train our student models to replicate. It is important to note that the kind of improvements regarding the use of the auxiliary model $\mathcal{T}$ (as discussed in Section 3.1) cannot be applied during KD pre-training, as KD provides the training signal for the student model by relying on the final output of $\mathcal{T}$. As such, KD pre-training requires strictly more compute compared to both MLM and MLSM.

### 4.2 Fine-tuning experiments

We primarily measure the quality of the models that we pre-train by quantifying their fine-tuning performance over a wide range of benchmark tasks. During this kind of assessment, we rely on the Corpus of Linguistic Acceptability (COLA; Warstadt et al., 2019), the CoNLL 2003 dataset for NER (Tjong Kim Sang & De Meulder, 2003), the MNLI natural language inference dataset Williams et al. (2018), the Microsoft Research Paraphrase Corpus (MRPC; Dolan & Brockett, 2005), the QNLI benchmark Rajpurkar et al. (2016); Wang et al. (2019b) datasets, Quora Question Pairs (QQP; Iyer et al., 2017), Recognizing Texutal Entailment (RTE; Dagan et al., 2006; Haim et al., 2006; Giampiccolo et al., 2007; Bentivogli et al., 2009), Stanford Sentiment Treebank (SST2; Socher et al., 2013), Semantic Textual Similarity (STSB; Cer et al., 2017) and the Word-in-Context (WiC; Pilehvar & Camacho-Collados, 2019) datasets.

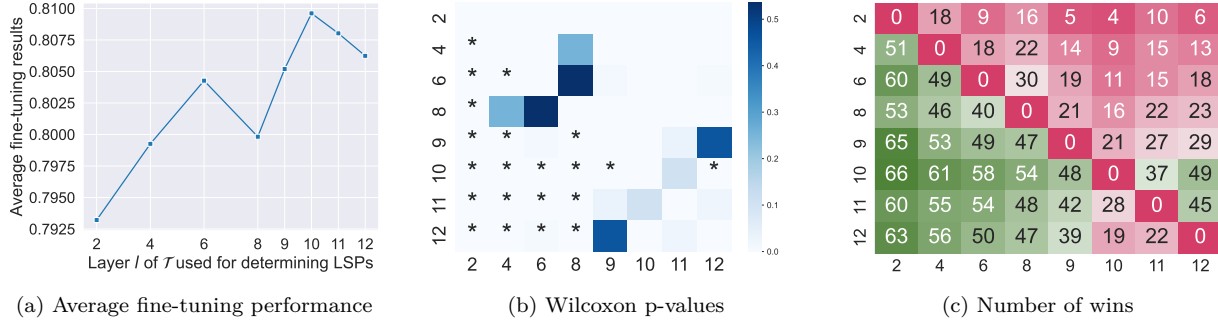

(a) Average fine-tuning performance    (b) Wilcoxon p-values    (c) Number of wins

Figure 3: The effects of using hidden states from different layers of the auxiliary model. (a) Aggregated fine-tuning performance averaged over 70 evaluations for each different choices of layer index. (b) p-values of the Wilcoxon signed rank test. A $*$ at the intersection of a row, column pair labeled with $(l_i, l_j)$ means that the MLSM model relying on layer $l_i$ of $\mathcal{T}$ performed significantly better ($p < 0.01$) than the MLSM model using layer $l_j$. (c) A value at position $(l_i, l_j)$ refers to the number of paired fine-tuning trials such that the MLSM model pre-trained over the LSPs based on layer $l_i$ of $\mathcal{T}$ scored higher.

As many of the datasets are part of the GLUE (Wang et al., 2019b) and SuperGLUE benchmarks (Wang et al., 2019a), where the labels of the test set are not available, we performed our evaluation on the development sets. We opted for frequently used hyperparameters for fine-tuning the datasets. That is, we used a learning rate of $2e-5$ with linear learning rate scheduling and a batch size of 32, performing 3 epochs. As the evaluation metric, we always report the fine-tuning performance after the last epoch.

We repeated every fine-tuning experiment 5 times and report the average performance of these trials with a differently initialized classification head in order to account for the potential variability in the fine-tuning performances of the individual experiments. We also perform the Wilcoxon signed rank test between pairs of differently pre-trained models, based on those pairs of fine-tuning experiments for which the classification heads of the models were initialized with the same weights. Performing the fine-tuning experiments for a model checkpoint took approximately 1 day on an NVIDIA A6000 GPU.

### 4.2.1  Experiments on the choice of $l$

Prior work on MLSM did not carefully evaluate the effects of using hidden vectors originating from other than the final layer of $\mathcal{T}$. This, however, potentially limited MLSM to reach its full potential, in light of prior evidence suggesting that the hidden representations of the last transformer block might not convey the most useful semantic information (Zhao et al., 2020). As discussed in Section 3.1, using earlier layers of $\mathcal{T}$ has the additional benefit of reducing the costs of pre-training, by making LSP calculation cheaper.

We were hence curious whether the utility of the LSPs can be improved by relying on hidden states from earlier layers of $\mathcal{T}$. As such, we went beyond the originally proposed strategy of using the final layer of $\mathcal{T}$ for obtaining LSPs, and trained a series of MLSM models while varying nothing else, but the layer of the transformer block of the auxiliary model $\mathcal{T}$ that we use for determining the LSPs.

We pre-trained a separate MLSM model while relying on the hidden states produced by any of the even indexed transformer blocks of $\mathcal{T}$. Starting with block 9, we also investigated the use of odd indexed blocks, as we originally expected the blocks near to the end of the network to produce more useful hidden states for determining the LSPs. In this section, we use the notation $\text{MLSM}_l$ for referring to the MLSM model that was pre-trained based on LSPs obtained from layer $l$ of $\mathcal{T}$.

Our evaluation suite consisted of 14 evaluations, each of which were repeated 5 times, resulting in 70 experiments per model checkpoints. We report the overall averaged results over these 70 fine-tuning scores conducted for each of our pre-trained MLSM models in Figure 3. A detailed breakdown on the individual task performances averaged over the 5 experiments per evaluation criterion can be found in Table 1.

Table 1: The detailed effects on the per-task averaged fine-tuning performance of MLSM models as a function of the layer of $\mathcal{T}$ used for determining the LSPs.

| task (metric) | Layer $l$ of the auxiliary model $\mathcal{T}$ used by MLSM$_l$ | | | | | | | |
|---|---|---|---|---|---|---|---|---|
| | 2 | 4 | 6 | 8 | 9 | 10 | 11 | 12 |
| CoNLL2003 (F1) | 0.946 | 0.947 | 0.948 | 0.950 | **0.951** | 0.949 | 0.949 | 0.950 |
| COLA (MCC) | 0.339 | 0.371 | 0.418 | 0.372 | 0.387 | 0.405 | **0.428** | 0.410 |
| MNLIm (accuracy) | 0.805 | 0.809 | 0.815 | 0.818 | 0.821 | **0.824** | 0.823 | 0.823 |
| MNLImm (accuracy) | 0.814 | 0.819 | 0.821 | 0.827 | 0.828 | 0.831 | **0.833** | 0.829 |
| MRPC (accuracy) | 0.843 | **0.864** | 0.861 | 0.837 | 0.859 | 0.860 | 0.859 | 0.851 |
| MRPC (F1) | 0.892 | **0.906** | 0.904 | 0.886 | 0.902 | 0.902 | 0.903 | 0.896 |
| QNLI (accuracy) | 0.881 | 0.883 | 0.888 | 0.898 | 0.891 | **0.902** | 0.898 | 0.895 |
| QQP (accuracy) | 0.897 | 0.899 | 0.901 | 0.903 | 0.901 | **0.903** | **0.903** | 0.902 |
| QQP (F1) | 0.861 | 0.864 | 0.865 | 0.867 | 0.865 | **0.868** | **0.868** | 0.867 |
| RTE (accuracy) | **0.608** | 0.585 | 0.588 | 0.584 | 0.601 | 0.601 | 0.593 | 0.593 |
| SST2 (accuracy) | 0.892 | 0.899 | 0.905 | 0.899 | 0.909 | **0.914** | 0.910 | 0.906 |
| STSB (pearson) | 0.845 | 0.844 | 0.851 | 0.857 | 0.858 | **0.859** | 0.857 | 0.856 |
| STSB (spearmanr) | 0.842 | 0.840 | 0.848 | 0.855 | 0.856 | **0.857** | 0.855 | 0.853 |
| WiC (accuracy) | 0.641 | 0.662 | 0.646 | 0.643 | 0.642 | **0.659** | 0.633 | 0.656 |
| Avg. | 0.793 | 0.799 | 0.804 | 0.800 | 0.805 | **0.810** | 0.808 | 0.806 |

As illustrated by Figure 3a – apart from a small relapse for using layer 8 hidden representations of $\mathcal{T}$ – there is a monotonic increase in the average fine-tuning performance of MLSM$_l$ models for $l \leq 10$, after which point there is a decline in the average fine-tuning results. Our results corroborate our hypothesis that using the last hidden states from $\mathcal{T}$ for determining the LSPs is sub-optimal.

We also calculated the Wilcoxon signed rank test between the 70 pairs of fine-tuning results per pre-trained model pairs, the outcome of which is summarized in Figure 3b. We put a * to those model pairs MLSM$_i$ and MLSM$_j$ indicated by row and column labels $i$ and $j$, such that the paired fine-tuning results of MLSM$_i$ were significantly better ($p < 0.01$) according to the Wilcoxon signed rank test compared to those of MLSM$_j$. The values in Figure 3c refer to the number of paired fine-tuning experiments for which MLSM$_i$ scored better than MLSM$_j$.

Table 1 and Figure 3 clearly indicate that choosing MLSM$_{10}$ over MLSM$_{12}$ should be preferred. MLSM$_{10}$ is not only cheaper to pre-train, but it also performs significantly better than MLSM$_{12}$. Even though there is no statistically significant difference between MLSM$_{10}$ and MLSM$_{11}$, pre-training the former is cheaper as it relies on an earlier layer of $\mathcal{T}$. For any other choice of investigated values for $l$, MLSM$_{10}$ performs significantly better than MLSM$_l$.

From a practical consideration, when using an $L$-layered auxiliary model $\mathcal{T}$, it is recommendable to pre-train MLSM$_l$ for a value of $l$ that is not equal but close to $L$. Such a choice not only reduces the pre-training costs compared to MLSM$_L$ (as the cost of determining LSPs is directly proportional to the number of layers kept from $\mathcal{T}$), but it is also likely to provide LSPs that provide more useful pre-training signal. For the above reasons, we conducted our remaining experiments with MLSM$_{10}$.

**Recipe for choosing $l$** Our previous analysis showed that relying on a layer other than the last one of the auxiliary model should be preferred both for saving computation and for improving the model transferability to downstream tasks via fine-tuning. At this current stage, it still remains a question if there is an efficient way to anticipate the effects of using a particular layer $l$ of the auxiliary model without the need to actually perform multiple MLSM pre-trainings with different choices of $l$.

To this end, we next propose and evaluate a cheap to calculate diagnostic with high predictive power towards the utility of a particular set of LSPs belonging to layer $l$ of the auxiliary model $\mathcal{T}$. This is useful, as it allows us to select $l$ in a more principled way. As the LSPs are meant to encode latent semantic information,

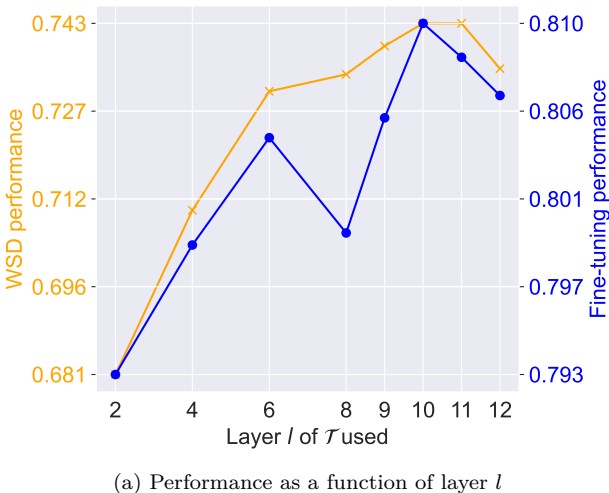
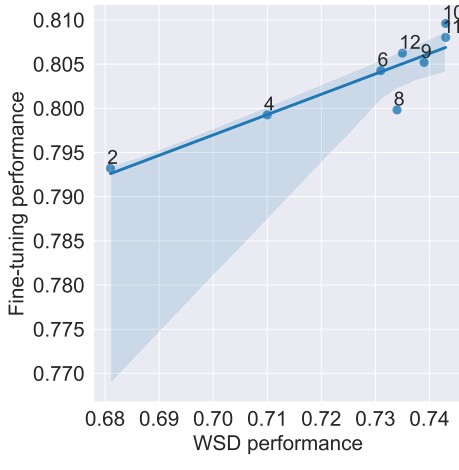

(a) Performance as a function of layer $l$

(b) Fine-tuning as a function of WSD performance

Figure 4: Comparison of the word sense disambiguation (WSD) and fine-tuning performance of the MLSM model that uses a given layer $l$ from the auxiliary model $\mathcal{T}$.

we argue that one way to assess their semantic content is via their use in word sense disambiguation (WSD). We anticipate that if the LSPs for layer $l$ of $\mathcal{T}$ perform better in the task of WSD, then we can expect those LSPs to be more useful when using them in MLSM pre-training.

We measured the utility of LSPs by evaluating them over the commonly used WSD benchmark of (Raganato et al., 2017). This benchmark includes the SENSEVAL-2 (Edmonds & Cotton, 2001), SENSEVAL-3 (Mihalcea et al., 2004), SemEval 2007 (Pradhan et al., 2007), SemEval 2013 (Navigli et al., 2013) and SemEval 2015 (Moro & Navigli, 2015) WSD datasets.

For this benchmark, the task is to determine the most likely WordNet (Fellbaum, 1998) sense for a total of 7253 ambiguous words in their context. For the WSD evaluation, we relied on the SemCor (Miller et al., 1993) sense-annotated dataset, consisting of 802,443 tokens with more than 28% (226,036) of those being sense-annotated with WordNet sensekeys.

In order to assess the utility of the LSPs, we first determined them for every sense-annotated token of SemCor. For some token $t$ with $\boldsymbol{h_t^{(l)}} \in \mathbb{R}^d$ denoting its contextual representation according to layer $l$ of $\mathcal{T}$, we determine $\boldsymbol{\alpha_t^{(l)}} \in \mathbb{R}^k$, i.e., its LSPs according to (2). In contrast to the latent semantic information, we denote with $\boldsymbol{s_t} \in \{0,1\}^{|S|}$ the explicit semantic description of a token according to the WordNet sense inventory. In $\boldsymbol{s_t}$, those indices are set to one for token $t$, that correspond to such a WordNet sense (out of the $|S|$ senses) that is relevant for the given token in its context. Based on both the explicit (WordNet-based) and the latent semantic representation of token $t$, we can determine $\boldsymbol{s_t^\mathsf{T} \alpha_t^{(l)}} \in \mathbb{R}^{|S| \times k}$ that we aggregate over all sense-annotated tokens to obtain $\Phi^{(l)} = \sum_t \boldsymbol{s_t^\mathsf{T} \alpha_t^{(l)}}$.

A particular entry of this matrix $\phi_{i,j}^{(l)}$ informs us about the strength of interaction between explicit semantic category $i$ and LSP $j$. We then transform $\Phi^{(l)}$ such that it stores the normalized pointwise mutual information (Bouma, 2009) between any pair of explicit and latent semantic categories. Then, for a query token $q$ with its LSP being given by $\boldsymbol{\alpha_q^{(l)}} \in \mathbb{R}^k$, we determine $\arg\max \Phi^{(l)} \boldsymbol{\alpha_q^{(l)}} \in \mathbb{R}^{|S|}$, which tells us the most likely WordNet synset that the token with the LSP profile $\boldsymbol{\alpha_q^{(l)}}$ belongs to.

Figure 4 reveals that the WSD performance achieved by a particular set of LSPs serve as a strong predictor towards the average downstream fine-tuning performance of MLSM pre-trained models that use the same set of LSPs as a pre-training signal. The benefit of performing WSD with the LSPs is that it can provide a cheap diagnostic, which allows us to choose a promising layer of the auxiliary model to use during MLSM. The predictive power of the WSD diagnostic towards the fine-tunability of the MLSM pre-trained model

using the same set of LSPs is supported by the fact that we measured correlation coefficients of 0.946 and 0.908 with corresponding p-values of 0.0003 and 0.0018 for the Spearman and Pearson correlation coefficients between the WSD and fine-tuning performances.

### 4.2.2 Experiments on the number of LSPs

A further previously unexplored question relates to the effects of choosing the number of latent semantic properties to consider during MLSM pre-training, i.e., the number of semantic atoms $k$, comprising the dictionary matrix $\boldsymbol{D}^{(l)} \in \mathbb{R}^{d \times k}$. The original implementation of MLSM relies on the choice $k = 3000$, however, it is not clear how sensitive the pre-training is to the choice of this hyperparameter.

To this end, we conducted 5 instances of MLSM pre-training under identical circumstances while varying only the value of $k$ over the values $\{768, 1536, 3072, 6144, 12288\}$. We report our results for each task in Table 2 and the average performance over all the evaluations conducted in Figure 5. Looking at the small differences in the figures of Table 2 and Figure 5a, we can conclude that MLSM is robust to the choice of the number of LSPs.

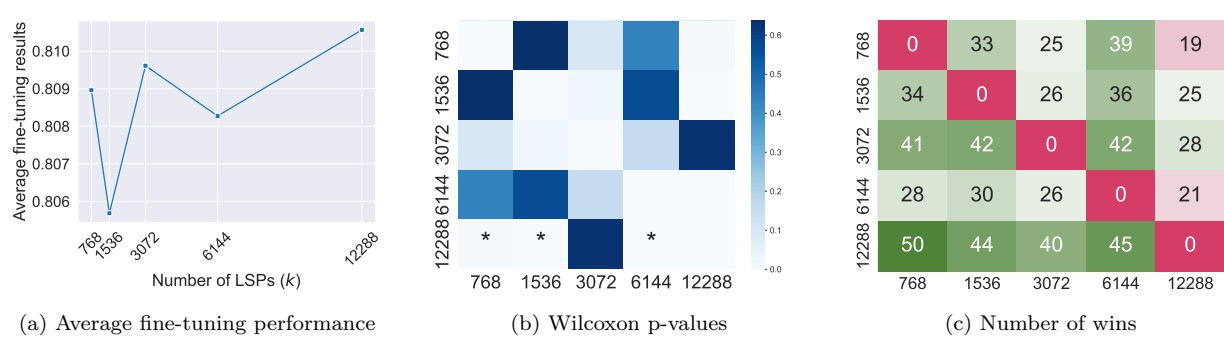

(a) Average fine-tuning performance     (b) Wilcoxon p-values     (c) Number of wins

Figure 5: The effects of using different number of LSPs. (a) Aggregated fine-tuning performance averaged over 70 evaluations for each different choices of $k$. (b) p-values of the Wilcoxon signed rank test. A $*$ at the intersection of a row, column pair labeled with $(k_i, k_j)$ means that the MLSM model relying on $k_i$ LSPs performed significantly better ($p < 0.01$) than the MLSM using $k_j$ LSPs. (c) A value at position $(k_i, k_j)$ refers to the number of paired fine-tuning trials such that the MLSM model $k_i$ LSPs scored higher.

This observation is further supported by the p-values of the Wilcoxon rank test that we present in 5b for the paired fine-tuning experiments belonging to a pair of MLSM models, the pre-training of which only differed in the number of LSPs employed.

Relying on orders more LSPs than previously recommended ($k \gg 3000$) not only results in a slight degradation of the fine-tuning performance of the pre-trained models (for $k = 6144$), but it also increases the computational need of pre-training, as the size of the model parameters responsible for making pre-training predictions is proportional to the number of LSPs (i.e., the unembedding parameters are of size $h \times k$). Based on the above, we conclude that the originally proposed number of latent properties was a reasonable choice, and we continue our experiments with $k = 3072$.

### 4.2.3 Analyzing pre-training efficiency

We next compare the efficiency of our modified MLSM model with those of MLM and KD pre-trained model variants. Our analysis covers both the investigation of the pre-training costs and the fine-tuning performance of these models.

**Computational need**   We include the cost of pre-training expressed in FLOPS over one batch of inputs with the different pre-training objectives and different MLSM variants in Table 3. We also break down the individual costs (i.e., the encoder and unembedding part) of the pre-trained and the auxiliary model. We calculated the FLOPS using the `calflops` Python package (https://pypi.org/project/calflops/).

Table 2: The detailed effects of using different number of LSPs ($k$).

| | Number of LSPs ($k$) | | | | |
|---|---|---|---|---|---|
| task (metric) | 768 | 1536 | 3072 | 6144 | 12288 |
| CoNLL2003 (F1) | 0.949 | 0.950 | 0.949 | 0.952 | 0.953 |
| COLA (MCC) | 0.407 | 0.382 | 0.405 | 0.392 | 0.393 |
| MNLIm (accuracy) | 0.824 | 0.826 | 0.824 | 0.822 | 0.823 |
| MNLImm (accuracy) | 0.830 | 0.831 | 0.831 | 0.831 | 0.831 |
| MRPC (accuracy) | 0.858 | 0.862 | 0.860 | 0.867 | 0.871 |
| MRPC (F1) | 0.901 | 0.904 | 0.902 | 0.907 | 0.909 |
| QNLI (accuracy) | 0.903 | 0.900 | 0.902 | 0.897 | 0.901 |
| QQP (accuracy) | 0.902 | 0.903 | 0.903 | 0.901 | 0.903 |
| QQP (F1) | 0.867 | 0.867 | 0.868 | 0.866 | 0.869 |
| RTE (accuracy) | 0.607 | 0.583 | 0.601 | 0.598 | 0.609 |
| SST2 (accuracy) | 0.912 | 0.911 | 0.914 | 0.908 | 0.915 |
| STSB (Pearson) | 0.855 | 0.857 | 0.859 | 0.855 | 0.859 |
| STSB (Spearmanr) | 0.852 | 0.855 | 0.857 | 0.853 | 0.857 |
| WiC (accuracy) | 0.658 | 0.648 | 0.659 | 0.665 | 0.657 |
| Avg. | 0.809 | 0.806 | 0.810 | 0.808 | 0.811 |

Table 3: The per batch costs of different pre-training paradigms and implementations. The naïve implementation of MLSM performs unembedding over $|V| + k$ symbols and a full forward pass over $\mathcal{T}$. In contrast, we perform unembedding over $|k|$ symbols and employ early exit at layer $l$ from $\mathcal{T}$ for MLSM$_l$.

| Pre-training | MLM | KD | naïve MLSM | MLSM$_{12}$ | MLSM$_{10}$ |
|---|---|---|---|---|---|
| trained encoder FLOPS/batch | 9.06e+12 | 9.06e+12 | 9.06e+12 | 9.06e+12 | 9.06e+12 |
| trained unembedding FLOPS/batch | 2.25e+12 | 2.25e+12 | 2.48e+12 | 0.29e+12 | 0.29e+12 |
| auxiliary encoder FLOPS/batch | — | 2.78e+12 | 2.78e+12 | 2.78e+12 | 2.32+e12 |
| auxiliary unembedding FLOPS/batch | — | 0.75e+12 | 0.75e+12 | — | — |
| Total FLOPS/batch | 1.13e+13 | 1.49e+13 | 1.50e+13 | 1.21e+13 | 1.16e+13 |
| FLOPS % | 100% | 131.9% | 132.7% | 107.1% | 102.7% |

We can see in Table 3 that KD and the naïve implementation of MLSM pre-training requires the highest FLOPS, i.e., the extra costs related to the use of the auxiliary model introduces more than +30% training cost over vanilla MLM pre-training. At the same time, the costs related to our improved MLSM implementation is on par with the use of MLM. This seemingly surprising phenomenon can partly be credited to the fact that it suffices to perform only forward computation in the auxiliary model, albeit this is the case in KD as well. Another source of efficiency when pre-training with MLSM is that we can early exit without the need to perform the calculation related to the final part of the model (also including a costly unembedding operation). Additionally, the unembedding in the trained model comes at a much reduced computation compared to those objectives that output symbols in the vocabulary space as opposed to the space of LSPs, since the size of the vocabulary has 10 times the number of LSPs .

**Pre-training dynamics** Over the course of pre-training, we created intermediate checkpoints of the models at their 10%, 25%, 50% and 100% readiness level, i.e., after performing 10000, 25000, 50000 and 100000 pre-training update steps. Figure 6 includes the average performance of different model variants as a function of the pre-training steps conducted. Figure 6a reveals the improved sample-efficiency of MLSM pre-training as the average fine-tuning performance of the MLSM model at its 50% readiness level is strictly higher than those of the alternatively pre-trained models at their 100% readiness.

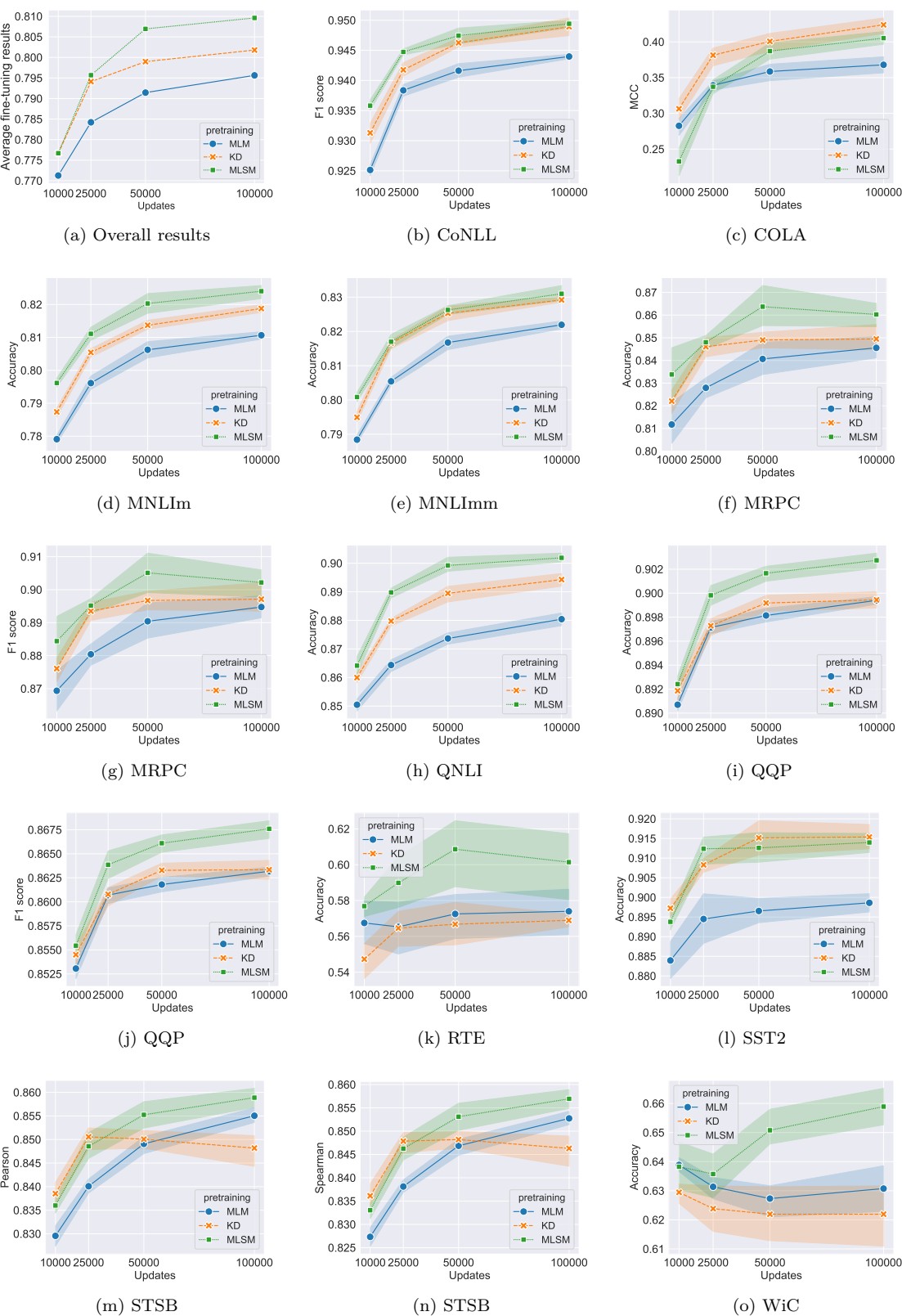

Figure 6: The average fine-tuning performance of differently pre-trained models as a function of their pre-training update steps.

Table 4: The detailed effects of using a multi-task learning pre-training with an MLM weight of $\kappa$.

| task (metric) | MLM weight $\kappa$ | | | | |
|---|---|---|---|---|---|
| | 0.0 | 0.25 | 0.5 | 1.0 | 2.0 |
| CoNLL2003 (F1) | 0.949 | 0.947 | 0.948 | **0.951** | 0.947 |
| COLA (MCC) | **0.405** | 0.382 | 0.392 | 0.399 | 0.363 |
| MNLIm (accuracy) | **0.824** | 0.822 | 0.822 | 0.819 | 0.818 |
| MNLImm (accuracy) | **0.831** | **0.831** | 0.827 | 0.828 | 0.826 |
| MRPC (accuracy) | 0.860 | 0.849 | 0.850 | **0.867** | 0.860 |
| MRPC (F1) | 0.902 | 0.895 | 0.896 | **0.907** | 0.902 |
| QNLI (accuracy) | **0.902** | 0.895 | 0.894 | 0.901 | 0.895 |
| QQP (accuracy) | **0.903** | 0.901 | 0.900 | 0.901 | 0.900 |
| QQP (F1) | **0.868** | 0.865 | 0.864 | 0.866 | 0.864 |
| RTE (accuracy) | 0.601 | 0.625 | 0.594 | **0.613** | 0.592 |
| SST2 (accuracy) | **0.914** | 0.908 | 0.912 | 0.912 | 0.906 |
| STSB (pearson) | **0.859** | 0.857 | 0.857 | 0.855 | 0.852 |
| STSB (spearmanr) | **0.857** | 0.855 | 0.854 | 0.853 | 0.851 |
| WiC (accuracy) | **0.659** | 0.649 | 0.644 | 0.637 | 0.636 |
| Avg. | **0.810** | 0.806 | 0.804 | 0.808 | 0.801 |

If comparison was made on the basis of FLOPS then the advantage of MLSM over KD was even more pronounced, while the comparison would practically be the same between MLSM and MLM (as KD has +30% extra compute cost over both MLM and MLSM pre-training, whereas MLM and MLSM have comparable computational need). Figure6 (b)–(o) provides further insights into the per evaluation setting performances of the differently pre-trained but otherwise identically fine-tuned models.

In Figure 7 we report the performance metrics of the individual experiments that we conducted by fine-tuning our models at their different readiness levels. In that figure, every marker covers a pair of fine-tuning results obtained by a pair of differently pre-trained models (for the same amount of updates) when initialized with the same set of task oriented classification parameters at the beginning of their fine-tuning. The identical initialization (and the fact that the batches came with identical contents and identical order) makes such fine-tuning experiments comparable with each other. Pairs of fine-tuning experiments for which the MLSM pre-trained model performed better are above the dashed line of the subplots.

We can see in Figure 7 that fine-tuning results obtained by the MLSM pre-trained model almost always reach or surpass those of the alternatively pre-trained models for all tasks and model readiness levels. COLA is the single notable exception, for which the fine-tuning results of alternatively pre-trained models is almost always better than that of MLSM models, which is something that we will focus on next.

### 4.3 Investigating the linguistic capabilities

Figure 6 and Figure 7 reveal that COLA is the single task where fine-tuning MLSM models could not reach the performance of alternatively pre-trained models. This can be attributed to the fact that the alternative pre-training objectives aim at predicting *actual word forms* (as opposed to semantic latents) which provides them increased transferability to such a task, where the goal is to decide on the linguistic acceptability of texts.

Indeed, linguistic acceptability is less related to semantics, and more about syntax and actual word forms, e.g., if a masked word was originally in its singular form, replacing it by its – semantically highly related – plural form will result in an agrammatic sentence. As an attempt to mitigate this shortcoming of MLSM, we additionally implemented a multi-task learning (MTL) approach, during which the MLSM and MLM objectives are jointly taken into account with hyperparameter $\kappa$ controlling the extent to which the MLM objective is considered in the final loss term, i.e., $\mathcal{L}_{MTL} = \mathcal{L}_{MLSM} + \kappa \mathcal{L}_{MLM}$.

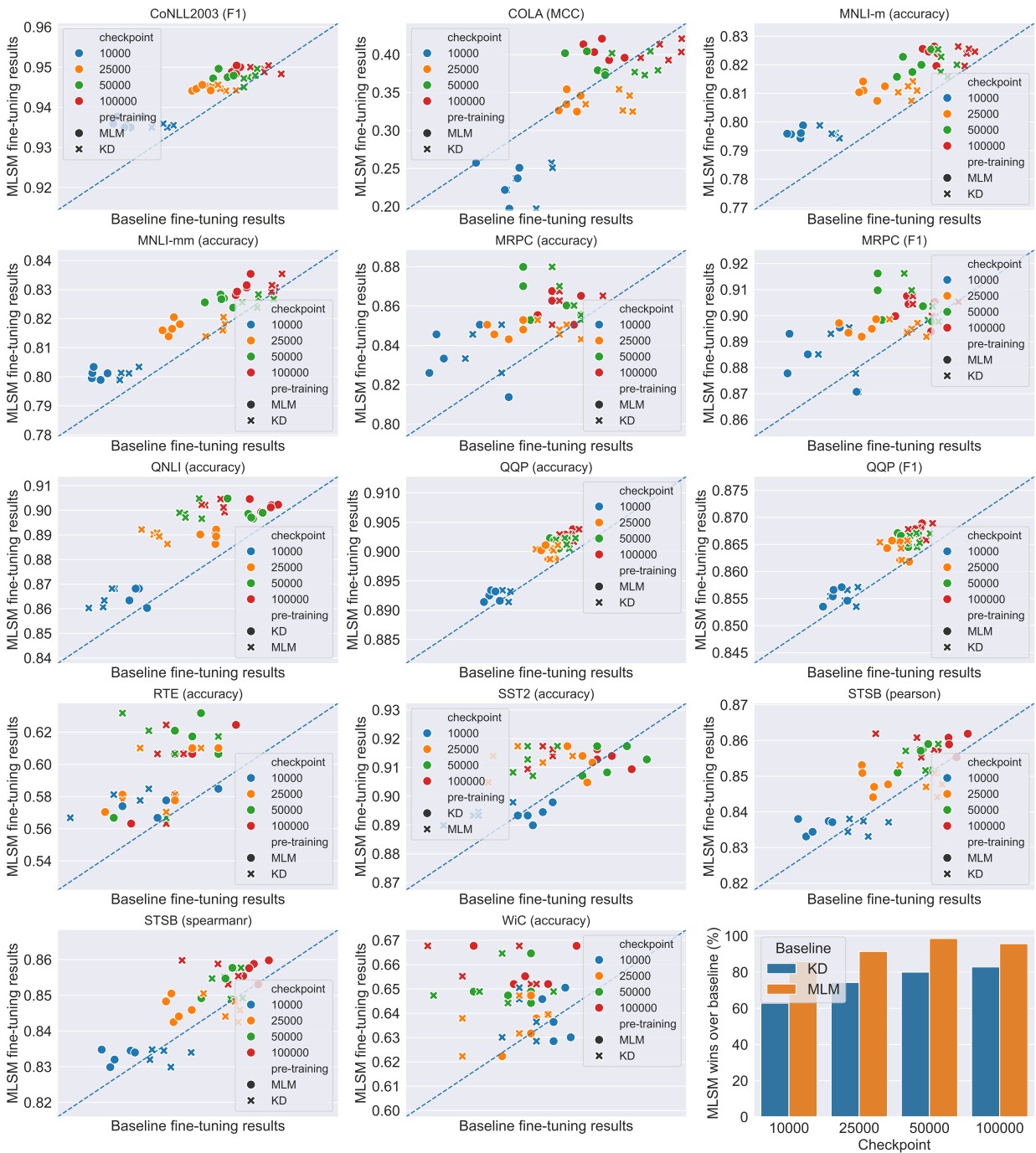

Figure 7: Scatterplot of paired fine-tuning experiments initialized with the same classification head. Colors indicate the checkpoint of the investigated models. Markers with circle and cross shapes are used to differentiate between comparisons against MLM and KD pre-trained models, respectively. Markers above the dashed diagonal line indicate fine-tuning experiments when the MLSM pre-trained model performed better than the alternatively pre-trained model variant at a given readiness level. The bar chart indicates the fraction of experiments such that the fine-tuning performance of the MLSM pre-trained model outperformed that of the alternatively pre-trained model variant for the different checkpoints.

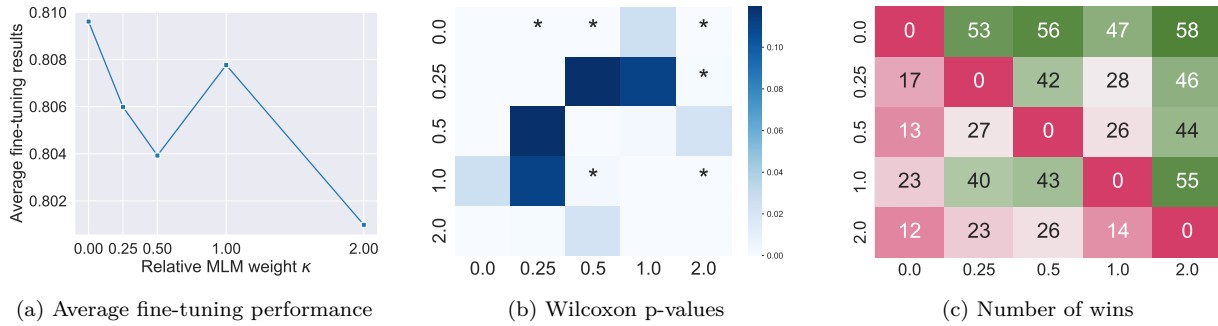

(a) Average fine-tuning performance      (b) Wilcoxon p-values      (c) Number of wins

Figure 8: The effects on the average fine-tuning performance of models jointly pre-trained with a combination of MLSM and MLM losses, $\kappa$ controlling for the extent to which the MLM loss is considered.

To this end, we trained and evaluated 4 additional MTL models with $\kappa \in \{0.25, 0.5, 1.0, 2.0\}$. MTL not only introduces an extra hyperparameter ($\kappa$) to account for, but we also need to sacrifice most of the efficiency improvements of MLSM pre-training, as unembedding has to be performed towards $|V| + k$ symbols (the additional $|V|$ being necessitated by the use of the MLM objective).

Based on our results included in Table 4 and Figure 8, we can conclude that the increased computational costs for MTL do not pay off. For all the investigated $\kappa > 0$ settings, we see worse performances compared to the $\kappa = 0$ case. Additionally, the differences are statistically significant according to the Wilcoxon signed rank test for all but the $\kappa = 1$ case. Even for that single case, the computational resources needed by the MTL model variant is substantially higher compared to the non-MTL ($\kappa = 0$) setting as discussed earlier. The general trend is that increasing the weight of MLM (hence decreasing the relative importance of MLSM) term in the loss makes fine-tuning performance to decline. This also implies that MLSM objective provides better downstream transferability. As such, we recommend against MTL pre-training.

### 4.3.1 Evaluation on BLiMP

There are arguably situations where the ability of a model to provide meaningful substitutes to a (masked) token position in an input sequence is beneficial. MLSM models, however, totally lack this capability, as they are not trained to reconstruct the exact identity of word forms, but they are capable of outputting the latent semantic properties of the tokens.

To this end, our final experiment investigates the extent to which we can predict the identity of actual masked tokens from the LSPs that are produced by an MLSM pre-trained model. Such an experiment also provides a way to assess the quality of the LSPs, i.e., in case we can accurately predict the actual masked word forms, then the LSPs likely encode useful semantic information.

As stated above, MLSM model cannot be meaningfully used for solving tasks that explicitly require predictions over the vocabulary of word forms of the model, such as it is the case with BLiMP (Warstadt et al., 2020), where models are evaluated by measuring the extent to which they assign higher pseudo log-likelihood (Salazar et al., 2020) score for linguistically appropriate token substitutions. Since the output space of MLSM is not tied to subword units, a direct evaluation of MLSM models would not be possible out of the box.

In order to still be able to evaluate MLSM pre-trained models for the BLiMP setting, we extended our model with a single linear layer on top of the outputs of the MLSM model, which can be viewed as a form of linear probing (LP). In this experiment, we froze all the model weights of the MLSM pre-trained models and added a single linear layer having $k \times |V|$ parameters for predicting token outputs based on the semantic latents produced by the MLSM model.

We kept the training of the single linear head deliberately short, i.e., we conducted 10,000 update steps altogether (whereas the MLSM model itself was trained for 10 times the update steps). The rational behind

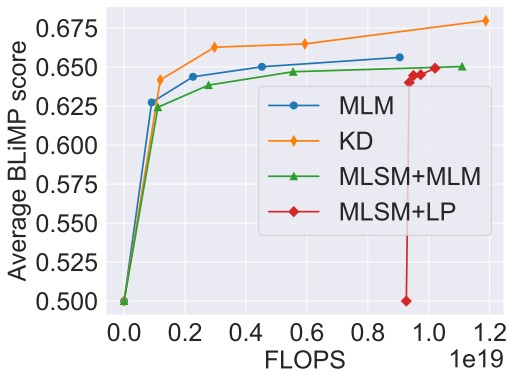

Figure 9: Average BLiMP evaluation results of the differently obtained model variants.

this was that if the LSPs learned during MLSM pre-training are useful and transfer to the task of masked language modeling, then a short training of the classifier head should suffice. Not only the training of the linear head took much shorter, but also most parts of the network did not require gradient computation, as we trained only a single linear layer of the network. As such, training the linear probe took no more than 3 GPU hours. Similar to our pre-training experiments, we created checkpoints at 10%, 25%, 50% and 100% readiness levels, meaning that our first linear probe required less than 20 minutes of extra compute.

Figure 9 contains our results on the BLiMP task for differently obtained models. We also include results of the jointly pre-trained MLSM+MLM model that corresponds to the kind of multi-task learning setting introduced in Section 4.3.1. We report results for the MTL model with $\kappa = 1$, as this is the MTL model that puts the largest emphasis on the MLM loss, something that is beneficial for a task like BLiMP.

Figure 9 includes the average BLiMP performance as a function of the amount of compute that was required for obtaining a given model variant. When reporting results for the MLSM model with the linear probe (MLSM+LP), we also include the costs of pre-training the MLSM model for which the extra linear head was added. This is the reason why the first marker for the MLSM+LP curve is not located at 0.0, but at the amount of computation that corresponds to the pre-training costs of the MLSM model itself. We can see from Figure 9 that we could extend the masked language modeling capabilities of our MLSM pre-trained model by using a negligible amount (less than 3 hours) of extra masked language modeling specific post-training.

This extra training did not affect the previously analyzed capabilities of our model, as the backbone weights were frozen. The BLiMP performance of the MLSM+LP model is on par with the MLM and MLSM+MLM models and only 2.5 points away from that of KD (but the KD model on the other hand is not able to output latent semantic properties, something that the MLSM+LP model is still capable of).

### 4.3.2 Scaled up pre-training experiments

Even though our primarily motivation is to provide such a pre-training paradigm that converges fast to a state that can perform well when being transferred to a downstream application – which is among the most desirable properties of encoder-based language models –, we also investigated the effects of MLSM when being employed over a much longer pre-training phase.

In order to do so, we performed the following modifications. Most importantly, we replaced our 3.4 billion token Wikipedia-based pre-training corpus with the Falcon RefinedWeb (Penedo et al., 2023) pre-training corpus included in Dolma v1.7 (Soldaini et al., 2024). We also increased the maximum sequence length and the amount of update steps by a factor of two (i.e., to 256 and 200,000, respectively) and the effective batch size by a factor of four (to 4096). These changes mean that instead of pre-training over approximately 13B tokens (4 epochs over the Wikipedia corpus), the total number of tokens seen during pre-training has been increased beyond 200B.

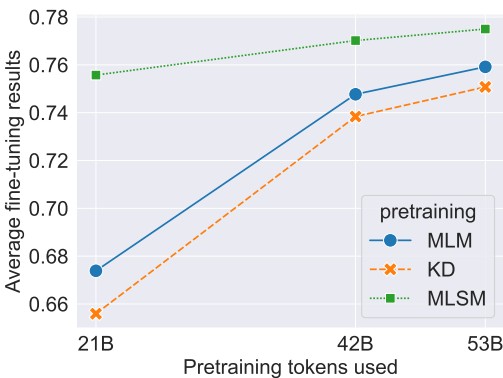

Figure 10: Average downstream fine-tuning performance of the differently pre-trained BERT models.

We also changed the model that we pre-trained from DeBERTa to BERT architecture. Using BERT was partially motivated from practicality as it is approximately 10% cheaper to train than DeBERTa. This was beneficial as our pre-training time have already prolonged substantially due to the longer training, i.e., from approximately 3 GPU days to 6 GPU weeks. Another benefit of changing the architecture was that this way we were able to test the generalization ability of the hyperparameters that worked well when pre-training a DeBERTa model.

Here, we report our (partial at the time of updating the manuscript) results that we obtained for fine-tuning the intermediate model checkpoints after 10%, 20% and 25% of pre-training. We evaluated the downstream transferability of the models over the same test suite as before and report the average performance in Figure 10. We can see that the same hyperparameters that worked for pre-training DeBERTa with MLSM also worked for the BERT architecture. Perhaps more importantly, we can also observe that the transfer capabilities of the MLSM pre-trained model is much favorable over the alternatively pre-trained model variants, i.e., the MLSM pre-trained model that accessed 21 billion pre-training token matches those of the alternatively pre-trained models that had access to 53 billion pre-training tokens.

## 5 Conclusions and future work

Our goal in this paper was to advance the understanding of masked latent semantic modeling in multiple important aspects. To this end, we set up different research questions and conducted carefully designed experiments to answer them. Our experiments have revealed multiple previously unexplored characteristics of MLSM pre-training. One of our important finding is that by choosing the layer of the auxiliary model that is used for determining the pre-training signal not to be the final one, we can improve the efficiency of pre-training both in terms of compute requirements and better fine-tunability. We also developed a cheap to compute diagnostic for predicting the expected utility of using different layers of the auxiliary model during MLSM pre-training, making the choice of the layer more principled and resulting in improved expected utility. Additionally, we have also verified via a special form of linear probing that the latent semantic features MLSM pre-trained models develop are meaningful towards performing actual masked word prediction task – something that MLSM pre-trained models are not capable at all otherwise.

Even though the focus of this work was to investigate text-only pre-training of encoder-based language models, we are optimistic that a similar kind of pre-training could be extended to vision models as well. A detailed analysis of employing the methodology of MLSM pre-training in the vision setting is beyond the scope of our current paper, however, we did assess the feasibility of such a pre-training from an intuitive point of view. For illustrating that the visual LSPs are meaningful from a human perspective – and therefore expected to provide potentially useful pre-training signal for vision models –, we determined the LSP of a CLIP vision encoder (Radford et al., 2021) over the training dataset of the SemEval 2023 shared task on Visual Word Sense Disambiguation (Raganato et al., 2023). We display in Figure 11 the top 3 images that had the highest non-zero activation towards four of the visual LSPs determined. We can see that the

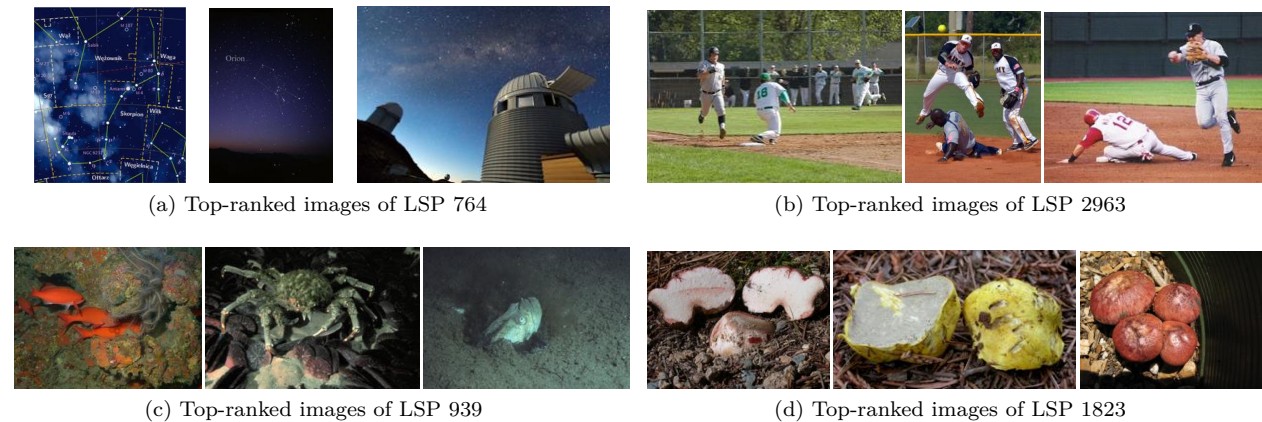

Figure 11: Top activating images of four LSPs that we trained on the SemEval 2023 dataset on Visual Word Sense Disambiguation (Raganato et al., 2023). LSPs correspond to human interpretable concepts. (a) astronomy, (b) baseball, (c) underwater life, (d) crops.

visual LSPs nicely correspond to human interpretable themes, which is something that we consider as an encouraging sign which hints that LSPs can provide useful training signal to vision models as well. Whether this hypothesis holds is something we wish to investigate in the future.

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
