# OpenReview forum: "Hitchhikers' Guide to Masked Latent Semantic Modeling"
_TMLR — Rejected by TMLR_

### Review · Reviewer_Lf8n · 2025-05-16

**Summary Of Contributions:**

Masked Latent Semantic Modeling (MLSM) was proposed by Berend, 2023.
This paper extends investigations on MLSM.

More specifically, it is tested:

* What is the optimal teacher layer to extract the latent semantic properties (LSP) (the sparse coding representations) to be used for the teacher-student MLSM training?

* What dimension of the sparse coding / amount of LSPs is optimal?

* Computational efficiency.

**Audience:**

No

**Broader Impact Concerns:**

(None)

**Claims And Evidence:**

No

**Requested Changes:**

Improve the abstract. Summarize the main findings.

Clarify Sec 3: It's not totally clear to me: Why are the sparse linear coefficients alpha l1-normalized? What does it mean to be l1-normalized? How is this enforced?

Clarify Sec 3 "MLSM pre-training then considers these sparse normalized distributions of latent semantic properties of the masked tokens as the desired target outputs" - I don't understand this.

The sparse normalized distributions of LSPs, that refers to the alpha? How are they normalized? They are a proper probability distribution, i.e.  alpha_i in [0,1] and sum_i alpha_i = 1? Is this the case? Why?

And then, how are they used as desired target outputs? Target for what exactly? The student does not predict the alphas, but it predicts the vocab, or not? This must be explained, as this is quite a crucial point.

Maybe I misunderstand the resulting model that is being trained with MLSM. I was assuming, in any case, it would have an output vocab V. Is this not true? If not, then that should be very clearly defined, how the model actually looks like. I only got the suspicion that it might not have such an output at all after reading other parts (e.g. in Sec 4.3, where it is said that the additional |V| is being necessitated (only) by the use of the MLM objective). So, this got me very confused. Also, what I don't exactly understand is, how is the resulting model actually used for the evaluations on the downstream tasks?

Clarify: What is |V| actually? (You say you use k = 3072 later. But how does this relate to |V|?)


Clarify Sec 3.1 "in the official implementation of MLSM, a full forward pass over T – including the calculation of the model logits": In the original MLSM, the calculation of the model logits is also not needed, right? (Please clarify that. It's not totally clear to me.) If the logits are needed, then I don't really understand MLSM, and that description should be improved. If the logits are not needed, then it doesn't really seem to be so relevant as putting this as an "improvement" now. I would rather call this a bugfix of an oversight in the original implementation. I would certainly also not refer to this as the "naïve" implementation. It's very misleading this way.


Clarify Sec 4.2.3: How did you compute the FLOPS?

Clarify Sec 4.2.3: In MLM, you don't have a teacher at all, but in MLSM, you have, right? You are saying, in MLSM_10, running the first 10 layers of a teacher model only adds 2.7% to the runtime? This cannot be. Or is this a really tiny teacher model compared to the student model? Or did you actually not count the teacher model at all in the FLOPS?

Clarify: Sec 4.2.3 "naïve implementation of MLSM pre-training" - What exactly does this mean? How does this differ to MLSM_{12}? What layer is used in the naïve implementation? It says it performs unembedding over |V | + k, but this is not really clear to me.

Extend experiment: Table 4 / Figure 5a, what happens for larger κ? From the graph, I would expect that the performance might increase even more?


"Even though MLSM pre-training has been shown to be more sample efficient than MLM" - I'm not sure this was really shown. In the original paper, it is argued from the training curve from pretraining, that in earlier steps, the performance is already better for MLSM compared to MLM. I don't think that you can conclude from this that it is more sample efficient. Sample efficiency would mean that MLM needs more samples (larger training corpus) to get the same performance as MLSM, or MLSM needs less samples (smaller training corpus) to get the same performance as MLM. This is not really shown. Also note, if such experiments are done, learning rate scheduling would need to be adjusted correctly in each case. Such experiments would maybe be some interesting extension to the presented work.


More stepping back: Motivate how the findings are maybe also interesting beyond MLSM.

**Strengths And Weaknesses:**

Strengths:

* Public code.

* Good evaluations.

* Comprehensive experiments.


Weaknesses:

* Abstract is quite short, and while it describes the topic of the paper, it doesn't say anything about the results.

* Many things are not clear to me. For example, MLSM is not totally clear. Those k extra labels are added to the vocab? How does the target prob distrib look like then during training? It's not really defined clearly. The MLSM training loss is never explicitly written down but only described in the text in a vague way. Also see the requested changes below for a list of things which should be clarified, or where the description should be improved.

* It is claimed that MLSM have better sample efficiency, but I'm not sure that this is really shown. See also in the requested changes part below on this.

* It is unclear to me how many individuals in the TMLR's audience are interested in findings on MLSM. So far, it seems that MLSM is used only by a single person (judging from all the 4 citations of the original MLSM paper), so my concern is that this number is very low or maybe even 0. I'm not sure whether the findings are useful beyond MLSM.

---

> ### Author Response · Authors · 2025-06-17
> **Thank you for the valuable feedback**
>
> We would like to thank the reviewer for the valuable feedback, which helped us in improving the clarity of the manuscript.
>
> Below are our answers to the remarks mentioned in the review.
>
> * Short abstract
>
> We have vastly rewritten the abstract which will explicitly contain our main results and methodological novelties.
>
> * ``things are not clear [...] Those k extra labels are added to the vocab? [...]''
>
> The original MLSM implementation added $k$ extra symbols, increasing the output space of the model by $k$ over the original vocabulary.
> This implementation, however, is wasteful (unless one is to implement multi-task learning over the joint objective of MLM and MLSM), as the logits responsible to the vocabulary units were discarded in the end.
> The improved implementation simply determines $k$ dimensional logits, whereas the naive implementation outputted a $|V|+k$ dimensional logit (the first $|V|$ dimensions of which were *not* used).
> The new version of the manuscript makes this difference more transparent.
>
> * ``It is claimed that MLSM have better sample efficiency, but I'm not sure that this is really shown.''
>
> We agree that characterizing MLSM as having improved sample efficiency might not be the most appropriate wording as it leaves out the fact that we are essentially referring to the improved performance of the fine-tuned models that can be achieved by MLSM checkpoints that are pre-trained shorter than other models.
> To this end, we will rather refer to the *improved fine-tuning transferability* of MLSM over alternative approaches in the new version of the manuscript as opposed to its better sample efficiency.
>
> Sample efficiency (in the regular sense) of the pre-training task is not that important after all when self-supervised training data can be easily collected (which is the case for languages for which large corpora can be collected).
> It is, however, an important aspect that the pre-training need not have to take as long when using MLSM, so that one can obtain a similar (or even better) downstream task-oriented fine-tuned model based on earlier checkpoints and/or allocating fewer pre-training steps.
>
> The below table illustrates that both the MLSM model trained for a total of 100K update steps at its 50% readiness and the MLSM model trained to 100% completion over 50K update steps perform on par with models pre-trained alternatively (with MLM or KD) to 100% completion over twice the updates.
>
> | | MLM   | KD | MLSM  | MLSM  |
> |-|-|-|-|-|
> | Total updates | 100K | 100K |100K | 50K |
> | Readiness | 100% | 100% | 50% | 100% |
> | Avg. fine-tuning performance | 0.796 | 0.802 | 0.807 | 0.797 |
>
> (We omit the per task performances of the different checkpoints for brevity, however, they behave similar to the overall average performances reported above.)
>
> * ``It is unclear to me how many individuals in the TMLR's audience are interested in findings on MLSM.''
>
> Estimating community engagement based on the count of citations at a given timestamp can be tricky.
> If someone had tried to judge the future interest towards neural networks in the early 2000s, they would probably not foresee their upcoming popularity.
>
> One potential reason why the community did not picked up on the idea that we build on could precisely be that the original paper lacked the kind of evaluations that we present in our manuscript.
> While we acknowledge that this is somewhat hypothetical, there are multiple concrete examples of recent papers [1,2,3] that exploit the idea of integrating latent semantics (or concepts in their terminology) into model pre-training, which we believe to go against the hypothesis that the amount of people interested in our work would be zero.
>
> While these recent work are focused on autoregressive models, the core idea is essentially the same, and having the kind of advantages integrated into the pre-training of encoder-based models is something that we believe to be of measurable community interest, especially that there has been a recent increased interest in encoder-based models [4,5,6].
>
> [1] Tack et al.: LLM Pretraining with Continuous Concepts. 2025. https://arxiv.org/abs/2502.08524
> [2] Chen et al.: Low-Rank Adapting Models for Sparse Autoencoders. In ICML 2025. https://arxiv.org/abs/2501.19406v1
> [3] Shani et al.: Towards Concept-Aware Large Language Models. In Findings of EMNLP 2023. https://aclanthology.org/2023.findings-emnlp.877/
> [4] Boizard et al.: EuroBERT: Scaling Multilingual Encoders for European Languages. 2025. https://arxiv.org/abs/2503.05500
> [5] Breton et al.: NeoBERT: A Next-Generation BERT. In TMLR. 2025. https://arxiv.org/abs/2502.19587
> [6] Warner et al.: Smarter, Better, Faster, Longer: A Modern Bidirectional Encoder for Fast, Memory Efficient, and Long Context Finetuning and Inference. In ACL 2025. https://arxiv.org/abs/2412.13663
>
> We are also grateful for the actionable list of requested changes.
> We implement them in the new version of the manuscript.

---

### Review · Reviewer_m4ph · 2025-05-30

**Summary Of Contributions:**

This paper investigated MLSM (masked latent semantic modeling). The authors argue that it serves as a more sample-efficient alternative to traditional masked language modeling for encoder-based LMs. Instead of predicting the exact identity of masked tokens, MLSM trains models to predict the latent semantic properties of these tokens, produced from a teacher model. They also provide empirical analysis of how to improve training efficiency of MLSM.

**Audience:**

Yes

**Claims And Evidence:**

No

**Requested Changes:**

Add more state-of-the-art baselines. It would be better to include scaling up experiments.

**Strengths And Weaknesses:**

Strengths:
1. MSLM is an under-explored area and such investigation will help the community to understand its advantages of different training objectives.
2. The authors provide a comprehensive empirical analysis which I really appreciate.

Weaknesses:
1. I can understand the advantages of MLSM compared to MLM, but currently autoregressive (AR) models dominate the field of large language models (LLMs). I’m curious—how does MLSM compare to AR models in terms of performance or efficiency? Are there any clear advantages?

2. Recently, diffusion LLMs have demonstrated promising performance. Could the authors provide an analysis comparing MLSM with diffusion-based LLMs, highlighting their respective strengths and weaknesses?

3. Regarding the experimental setup, I have a major concern: under the current landscape of large-scale language models, the settings used in this paper seem overly simplistic. For example, as mentioned in Section 4.1, the model is trained with a maximum length of 128 and on 13B tokens. In contrast, modern pre-training efforts (e.g., LLaMA, Qwen, or even proprietary models) typically involve training on trillions of tokens with a maximum sequence length of at least 2K. Therefore, I’m not sure whether the findings of this paper are directly applicable to real-world scenarios, or whether the observed effects would still hold when scaled up.

4. As for the baselines, the authors mainly compare MLSM with KD, MLM and similar methods, but there is a lack of comparison with current state-of-the-art large language models. Even within the encoder-only paradigm, comparisons to stronger or more recent baselines are missing.

5. Finally, there are issues with repetitive language and a lack of clarity in structure throughout the paper.
In page 2, "our approach, however" appears twice.

> Our approach, however, differs from these earlier works, which primarily focused on architectural optimizations and design choices within the traditional pre-training paradigm. Our approach, however, differs from these earlier works, as they were focusing on architectural speedups and design choices of encoder models trained with the traditional pre-training paradigm, whereas we focus on modification of the learning objective in order to make pre-training more sample efficient and better aligned with human perception.

---

> ### Author Response · Authors · 2025-06-17
> **Thank you for the feedback**
>
> We would like to thank the reviewer for the useful comments for improving our paper.
>
> Below, we response to the remarks of the review.
>
>  >   I can understand the advantages of MLSM compared to MLM, but currently autoregressive (AR) models dominate the field of large language models (LLMs). I’m curious—how does MLSM compare to AR models in terms of performance or efficiency? Are there any clear advantages?
>
> AR models are indeed popular, but there is still considerable community interest in encoder-only models as indicated by recent research efforts such as [ModernBERT](https://arxiv.org/pdf/2505.03574), [NeoBERT](https://arxiv.org/abs/2502.19587) or [EuroBERT](https://arxiv.org/abs/2503.05500).
> That is not to say that encoder models are superior to AR models, but they can provide complementary benefits and improved encoders still have a place in the current ecosystem.
>
> The ModernBERT authors did a great job in emphasizing the [importance of encoder based models](https://huggingface.co/blog/modernbert), we only highlight some of the reasons they mention why encoder-based models are still of interest: they can support generative models in various ways (e.g. in RAG setting) and they have reduced inference costs which can be highly desirable in latency-sensitive production environments.
> Currently, BERT is still one of the most frequently downloaded model on HuggingFace hub, and we believe that encoder-only models would stay with us for the foreseeable future.
>
> > under the current landscape of large-scale language models, the settings used in this paper seem overly simplistic. For example, as mentioned in Section 4.1, the model is trained with a maximum length of 128 and on 13B tokens. In contrast, modern pre-training efforts (e.g., LLaMA, Qwen, or even proprietary models) typically involve training on trillions of tokens with a maximum sequence length of at least 2K. Therefore, I’m not sure whether the findings of this paper> are directly applicable to real-world scenarios, or whether the observed effects would still hold when scaled up.
>
> The recent [LlamaFirewall](https://arxiv.org/abs/2505.03574) paper from Meta describes highly latency-critical use cases, in which they strive for model decision in the order of milliseconds.
> In particular they write that ''PromptGuard 2 is built using BERT-based architectures, [...] including mDeBERTa-base (86M parameters) and DeBERTa-xsmall (22M parameters)'', highlighting that there is still real-world need of small models (in as important applications as jailbreak detection).
>
> At the same time, we agree that scaling up pre-training can gain additional insights, to which end we started pre-training a BERT-base model with double the context size, and perhaps more importantly, over 200B unique tokens (as opposed to the 3.4B tokens of our current pre-training corpus).
> Pre-training has reached 25% by now, and the benefits of the model pre-trained with the MLSM objective seem to be at least as pronounced (if not more) under this setting. The below table contains the average fine-tuning performance of differently pre-trained models at their different readiness levels.
>
> |    | readiness |    |
> |------|--------|------|
> |    | 10%    | 25%   |
> | MLM  | 0.674| 0.759 |
> | KD   | 0.656  | 0.751 |
> | MLSM | 0.756 | 0.775 |
>
> > Recently, diffusion LLMs have demonstrated promising performance.
>
> We wanted to alter nothing else, but the pre-training objective between our baseline and proposed model variants, as this ensures that the differences that we see come only from the change of the objective function.
> In order to factor our any potential confounds (e.g. model capacity, architectural differences), we were relying over the same backbone architecture throughout our experiments for maximizing comparability between models.
>
> >  there is a lack of comparison with current state-of-the-art large language models. Even within the encoder-only paradigm, comparisons to stronger or more recent baselines are missing.
>
> We decided to go for the DeBERTa backbone architecture, which is generally considered as one of the strongest performing model variant among encoder-based models.
> Comparing with alternative baseline approaches, in which there are further differences other than the objective function, it would be difficult to assess where do the performance differences come from.
> To this end, we kept the architecture fixed throughout our experiments, as this ensured that the differences that we see come exclusively from the modification of the pre-training objective.
>
> >  Finally, there are issues with repetitive language
>
> Thank you for pointing this out, we have improved the manuscript in this respect.

---

### Review · Reviewer_jaDw · 2025-06-03

**Summary Of Contributions:**

This paper investigates Masked Latent Semantic Modeling (MLSM), a pre-training objective proposed as a more sample-efficient alternative to Masked Language Modeling (MLM) for encoder-decoder language models. Instead of predicting the exact masked word, MLSM trains models to output a context-sensitive semantic characterization of masked words, represented as latent semantic properties (LSPs). These LSPs are derived in an unsupervised manner from an auxiliary teacher model's hidden representations using sparse coding via dictionary learning. In this manuscript, the authors identify and evaluate previously unexplored aspects of MLSM pre-training. Their main research questions focus on:

* Improving the efficiency of MLSM.
* The effects of determining LSPs from different layers of the auxiliary model.
* The effects of using a different number of LSPs (i.e., the dictionary size k).
* Testing and improving the linguistic capabilities of MLSM pre-trained models.

**Audience:**

Yes

**Broader Impact Concerns:**

None.

**Claims And Evidence:**

Yes

**Requested Changes:**

* The authors are encouraged to more explicitly articulate the conceptual or methodological contributions beyond hyperparameter tuning.
* The finding that layer 10 of BERT-base yields the most effective LSPs is interesting, but may be architecture-specific. Please clarify whether similar trends were observed (or are expected) in other architectures.
* I am curious whether the core idea of MLSM—predicting context-sensitive latent semantics rather than discrete labels—could be adapted to masked visual modeling.

**Strengths And Weaknesses:**

S:

+ The paper methodically investigates key hyperparameters and design choices for MLSM (teacher layer l, number of LSPs k, efficiency improvements) that were not deeply covered in the original proposal.
+ The study translates its findings into clear, actionable recommendations for practitioners interested in using MLSM (e.g., choice of teacher layer, number of LSPs, discouraging MTL with MLM ).

W:

- While the paper provides a thorough empirical investigation and offers valuable practical insights, its primary contribution could be viewed as an extensive exploration and characterization of an existing method (MLSM by Berend, 2023 ). A significant portion of the research focuses on understanding the impact of different configurations and hyperparameters, such as the choice of the auxiliary model's layer (l) for LSP extraction  and the number of LSPs (k). Although these are important for the practical application and optimization of MLSM, the work primarily offers insights into "how to best use" an existing technique rather than introducing fundamentally new algorithms, architectures, or theoretical frameworks. This focus on parameter tuning and empirical characterization, while useful, might be seen as an incremental advancement rather than a breakthrough in terms of core methodological novelty.
- The conclusion that layer 10 (of a 12-layer BERT-base model ) is optimal for LSP extraction is interesting. However, this finding might be specific to the chosen auxiliary model (BERT-base). It would be beneficial to discuss or ideally show if this trend holds for different teacher model architectures or sizes. The paper states "prior evidence suggesting that the hidden representations of the last transformer block might not convey the most useful semantic information", but a deeper hypothesis for why layer 10 performs best in this specific MLSM context would be insightful.
- As a researcher coming from the CV community with limited background in NLP pre-training paradigms, I wonder whether the core idea behind MLSM—i.e., predicting latent semantic properties instead of discrete token identities—could be applicable or beneficial in vision-language models. If MLSM-like objectives can also be applied to visual pre-training (e.g., learning sparse semantic representations of masked image patches), this would significantly broaden the impact and conceptual significance of the method beyond NLP.

---

> ### Author Response · Authors · 2025-06-19
> **Author response**
>
> We would like to thank the reviewer for taking the time to review our submission and the valuable feedback provided.
>
> We reflect to the most important points of the review below.
>
> > The conclusion that layer 10 (of a 12-layer BERT-base model ) is optimal for LSP extraction is interesting [...]  but a deeper hypothesis for why layer 10 performs best in this specific MLSM context would be insightful
>
> We extended our analysis by employing a measure that evaluates the LSPs extracted from a particular layer of the auxiliary model in the task of word sense disambiguation.
> Our hypothesis was that good LSPs are also useful in differentiating between word senses and LSPs that are useful for sense disambiguation are expected to also be useful when used during pre-training.
> This diagnostic measure is orders of magnitude cheaper to compute than performing the entire pre-training and it turns out that this metric correlates strongly with the downstream fine-tuning performance of the MLSM model that uses the LSPs as a training signal from the given layer of the auxiliary model.
> It is true that this diagnostic metric do no always peak at layer 10 for all models (we tested it for DeBERTa, RoBERTa and BERT), but it is true that the final layer never reached the best diagnostic performance.
> Also, with the help of this cheap to compute diagnostic, one can choose the layer to obtain the LSPs from in a more principled and auxiliary model specific way.
>
> > thorough empirical investigation vs. with methodological novelty
>
> Inspired by the previous remark of the reviewer, we have tested the novel hypothesis that the utility of LSPs in pre-training can be predicted by the utility of the LSPs in solving word sense disambiguation.
> We were able to support this assumption, and since the assessment of the word sense disambiguation capability of LSPs is much cheaper than performing an entire pre-training, this offers a novel and more principled way of deciding which layer of the auxiliary model to determine the LSPs from during MLSM pre-training.
>
> > I wonder whether the core idea behind MLSM—i.e., predicting latent semantic properties instead of discrete token identities—could be applicable or beneficial in vision-language models
>
> While pre-training vision transformer models was beyond the scope and computational budget of our research, we anticipate MLSM-style pre-training to work for vision transformers as well.
> We find the proper investigation whether this is indeed the case as an interesting future research direction that we add to the conclusion of our manuscript.
>
> Even though we did not pre-train vision models, we conducted a qualitative analysis of the visual LSPs that we obtained from a CLIP model, which can potentially serve as an auxiliary model when being used in visual MLSM pre-training.
> Our qualitative analysis revealed that images that share a visual LSP are often highly related to each other, which we regard as a promising sign towards the applicability of MLSM pre-training in the visual setting as well.

---

### Decision · Action_Editor_2DRN · 2025-08-01

**Recommendation:** Reject

**Audience:**

No

**Audience Explanation:**

The biggest concern raised by the reviewers is that the paper's topic is very narrow -- purely focusing on MLSM, which the reviewers found to be a quite uncommon objective in the literature (multiple reviewers pointed this out in the final recommendation).
> I am not sure if the TMLR's audience would be interested in this direction... only a single person was using and working on it.

> I'm still not sure that the audience is maybe non-existing here.

Overall, there is not sufficient support from the reviewers to accept this submission in its current form.

While we do not discourage submissions that fall outside of mainstream methods, this submission seems to really suffer from a lack of audience at TMLR. To mitigate this concern, the authors should consider how the findings/insights in this paper can (1) transfer to autoregressive LM training which is the common paradigm in pretraining, and (2) generalize to MLM training which is a much more common objective to build encoder models.

**Claims And Evidence:**

Yes

**Claims Explanation:**

This paper studies Masked Latent Semantic Modeling (MLSM) which is an MLM-style pretraining objective that reconstructs latent semantic properties (LSPs). Through a series of empirical studies, this work offers several practical guidelines to improve the efficiency of MLSM pretraining. Overall, the reviewers generally found the empirical investigation to be thorough and solid, thus the claims are well-supported by the empirical results.

**Resubmission Of Major Revision:**

The authors may consider submitting a major revision at a later time.